# Topographical and Ultrastructural Evaluation of Titanium Plates Coated with PLGA, Chitosan, and/or Meropenem: An In Vitro Study

**DOI:** 10.3390/dj10120220

**Published:** 2022-11-26

**Authors:** Mohammad Al-Qubaisey, Rita Khounganian, Abdulhakim Al-Badah, Raisuddin Ali

**Affiliations:** 1Department of Dentistry, Riyadh 2nd Health Cluster, P.O. Box 60169, Riyadh 11545, Saudi Arabia; 2Department of Oral Medicine and Diagnostic Sciences, College of Dentistry, King Saud University, P.O. Box 60169, Riyadh 11545, Saudi Arabia; 3Microbiology Laboratory, College of Dentistry, King Saud University, P.O. Box 60169, Riyadh 11545, Saudi Arabia; 4Department of Pharmaceutics, College of Pharmacy, King Saud University, P.O. Box 60169, Riyadh 11545, Saudi Arabia

**Keywords:** titanium plates, scanning electron microscope, ultrastructural topography, chitosan, PLGA, meropenem, *Staphylococcus aureus*, *Pseudomonas aeruginosa*

## Abstract

The present investigation was undertaken to evaluate the topographical and ultrastructural architecture of titanium plates coated with polylactic co-glycolic acid (PLGA), chitosan (CH), and/or meropenem (MEM) with or without *Staphylococcus aureus* (*SA*) or *Pseudomonas aeruginosa* (*PA*) bacteria. Single-hole segments of 0.4 mm thick, low-profile titanium plates were spray coated using an airbrush with polymeric carriers (PLGA or CH) loaded with MEM, in addition to the negative control group (uncoated titanium plates). The coated plates and the negative control group were subjected to bacterial biofilms through a cultivation process while being slowly stirred at 20 rpm for 24 h. The samples were fixed and processed for scanning electron microscopic study at 5, 10, and 20 k magnification. The data were statistically analyzed to compare within and between the different materials. Coating titanium plates with PLGA or CH with MEM appeared to enhance bacterial inhibition over uncoated plates, hindering biofilm formation and preventing bacterial proliferation. In the staphylococcus aureus group, the highest bacterial count was observed in the uncoated plates, whereas the lowest count was detected in meropenem-PLGA, followed by PLGA, chitosan, meropenem, and meropenem-chitosan, respectively. On the other hand, the *Pseudomonas aeruginosa* group with the uncoated plates had the highest bacterial count, whereas the lowest bacterial count was found related to CH, followed by PLGA, MP, MC, and MEM, respectively.

## 1. Introduction

Microplates were initially introduced in the early 1970s for the fixing of facial fractures. Later, Haerle et al. popularized the use of microplates in orthognathic surgery and craniomaxillofacial injuries [1]. These plates were made from different alloys. However, because of their susceptibility to corrosion and low biocompatibility, stainless steel and other metal alloys have since fallen out of favor and have been replaced by titanium [2]. Owing to its superior strength-to-weight ratio, great corrosion resistance, and biocompatibility, titanium and its alloys are optimal for biomedical applications [3].

Currently, it is estimated that 10–12% of plates need to be removed due to infection, exposure, discomfort, or pain [4]. Smith et al., states that titanium plates and screws have a 10% failure rate [5].

Patients fitted with surgical titanium plates were reported to continue having complaints due to various reasons, for decades following surgery [5]. In addition, infections due to microplates are challenging to treat by conventional methods, such as the enteral and parenteral routes which are usually effective for the delivery of small molecule drugs to other infection sites [6]. 

Recent developments in biomacromolecular treatments have increased the need for innovative drug delivery methods because unbound macromolecules often have a short half-life and a proclivity to breakdown and denature in physiological environments [7]. Given their advantages over traditional drug delivery systems, polymeric nanoparticles, especially those made with biodegradable polymers, are suitable carriers of macromolecular pharmaceuticals [8].

The aim of coating titanium plates with copolymeric carriers augmented with antimicrobial agents is to decrease the possibility of infection to avoid the need for additional procedures. Polylactic-co-glycolic acid (PLGA) is one of the most common polymers for biological applications and drug delivery. It is a copolymer of lactic acid and glycolic acid that is biodegradable and biocompatible, with more efficiency than previous materials for sustained drug release. It is hydrolyzed in the body to produce biodegradable metabolites, primarily lactic acid and glycolic acid [9].

Chitosan (CH) is extracted from the shells of marine arthropods. CH has many advantages in biomedical applications, including biocompatibility and natural biodegradability, which bestow nontoxic and noninflammatory qualities on its degradation products. Being soluble in water, it has numerous applications that can alter its form and structure to produce films and scaffolds [10]. Meropenem (MEM) is part of the carbapenem subclass of the β lactam class of antibiotics with an exceptionally broad spectrum of activity [11]. 

To our knowledge, the literature lacks information regarding PLGA, or chitosan augmented with meropenem, to enhance the titanium plates used in oral and maxillofacial surgery. As a consequence, the present research was undertaken to verify and investigate the application and efficacy of PLGA, or CH enhanced with meropenem as a coating for titanium plates to prevent bacterial adhesion and colonization. As a prophylactic against infection in routine maxillofacial surgeries, particularly in reconstruction cases when the patient may have numerous comorbidities, these redesigned titanium plates are intended to replace their present counterparts. Moreover, this might reduce the chances of subjecting the patients to further surgeries and might reduce the burden upon the health care system.

In this study, the ultrastructural topography of coated titanium plates before and after exposure to two types of bacteria (Gram + ve *Staphylococcus aureus* [*SA*] and Gram -ve *Pseudomonas aeruginosa* [*PA*]) were evaluated using a scanning electron microscope to study the copolymer layer characteristics over the surface of the plate with or without bacterial adherence.

## 2. Materials and Methods

### 2.1. Sample Preparation

The present study was carried out on 54 single-hole segments of 0.4 mm thick, low-profile MatrixMIDFACE titanium plates (DePuy Synthes, Warsaw, IN, USA).

Coating materials were divided into two categories:A.Carriers: composed of copolymers PLGA and CHB.Antimicrobial agent: MEM

The following coating materials—PLGA or CH and MEM—were investigated using 1% *w*/*v* in acetone (Sigma-Aldrich, St. Louis, MI, USA) of each solution. They were sprayed onto the entire surface of the plates using an airbrush system (Harder & Steenbeck, Oststeinbek, Germany) [12]. The plates were left to air-dry overnight at room temperature in the safety cabinet. The coating procedure was carried out accordingly:

Solution 1: 25 mg PLGA and 5 mg meropenem in 2.5 mL acetone (MP)

Solution 2: 25 mg chitosan and 5 mg meropenem in 2.5 mL acetone (MC)

Solution 3: 5 mg meropenem in 2.5 mL acetone (MEM)

Solution 4: 25 mg PLGA in 2.5 mL acetone (PLGA)

Solution 5: 25 mg chitosan in 2.5 mL acetone (CH)

Each plate was weighed before and after spraying to ensure the final modified plates contained 5 mg of MEM and 25 mg of the polymer [13].

### 2.2. Bacteria Strains

Gram-positive aerobic (*Staphylococcus aureus* “SA” ATCC^®^ # 25923) and Gram-negative aerobic (*Pseudomonas aeruginosa* “PA” ATCC^®^# 27853) bacterial strains were selected for this research. Bacterial adhesion, proliferation, and growth, along with the biofilm formation, were studied accordingly.

### 2.3. Bacteria Cultivation

Tryptic soy broth (TSB; Sigma Aldrich, St. Louis, MI, USA) was used to cultivate the two types of bacteria (*SA* and *PA*). The cultivation took place overnight in an incubator set at 37 °C. The following day, the culture was diluted to 1:50 in fresh TSB and incubated for 3 h at 37 °C to achieve the logarithmic growth phase. A total of 25 μL (equal to 50 McFarland) of the bacteria culture was injected into 18 wells on six lines of a flat-bottomed 96-well microtiter polystyrene plate. One plate was used for each type of bacteria. One segment of each coated plate was placed into the 18 wells with the bacteria solution, including the uncoated negative control group. The polystyrene plate was gently spun at 20 rpm for 24 h and kept at 37 °C to generate biofilms [14,15]. 

### 2.4. Sample Preparation for Scanning Electron Microscope “SEM”

Each titanium plate segment was removed from the polystyrene plate using sterile forceps and gently washed with phosphate buffered saline (PBS; Sigma Aldrich, St. Louis, MI, USA) to remove planktonic bacterial cells. The plates were immersed for 4 h in 25% glutaraldehyde solution (Sigma Aldrich, St. Louis, MI, USA), then rinsed three times with PBS. The plates were dehydrated in ethanol solutions of increasing concentrations (50, 70, 80, 90, and 100%) for 5 min at each concentration. Samples were stored in 100% ethanol at 4 °C until examination time [16].

### 2.5. SEM Protocol

The processed samples were affixed to the stubs with double-sided carbon conductivity tape. A thin layer of gold was sprayed onto the samples using a Quorum sputter coater (Q150RS, Laughton, East Sussex, UK) under a vacuum in an argon atmosphere for 1 min. The biofilm adhesion to the titanium plates surface was examined at an accelerating voltage of 10 kV under a vacuum 4 × 10^6^ bar using Zeiss (EVO-LS10, Jena, Germany). The images were captured using Smart SEM software, version 5.05 (Zeiss, Jena, Germany) [16,17]. 

### 2.6. SEM Image Analysis

The bacteria were counted using ImageJ software (version 1.53, MD, Bethesda, USA), and the plates without bacteria were qualitatively analyzed [18].

### 2.7. Statistical Analysis

The collected data were analyzed using the Statistical Package for the Social Sciences software version 26.0 (IBM Inc., Chicago, IL, USA). Descriptive statistics (mean and standard deviation) were used to express all quantitative variables. Kruskal–Wallis nonparametric test was performed to compare the quantitative data obtained from the SEM imaging of the samples because normality was not satisfied. A Mann–Whitney U test was used to compare within and between the different materials. One examiner carried out all assessments and repeated them three times to confirm reproducibility and reliability. The results were considered statistically significant when *p* ≤ 0.05.

## 3. Results

### 3.1. Qualitative Assessment of Coated Plates without Bacteria

SEM analysis of the coated titanium plates was undertaken at 5, 10, and 20 k magnification. PLGA (Figure 1A,B) revealed a heterogeneous, smooth, and dense surface with shallow indentations. Scanty, small impurities were noticed at higher magnification (20 k, Figure 1C). MP-coated plates (Figure 1D,E) revealed a heterogeneous, smooth, and dense surface with wider, more shallow indentations than PLGA. Small, irregular-sized impurities were noticed as magnification increased to 20 k (Figure 1F). CH-coated plates (Figure 1G,H) revealed a heterogeneous, vastly irregular, and dense surface with more variably sized shallow indentations than PLGA. Variably sized impurities were noticed as magnification increased to 20 k (Figure 1I). MC-coated plates (Figure 1J,K,L) demonstrated an imperfect rough surface that was filled with irregularities, presenting as shallow grooves, elevations, and depressions of different sizes and shapes. MEM-coated plates (Figure 1M,N,O) showed a similar presentation to MC, except the indentations were wider, and the irregular particles were smaller. The uncoated plates (-ve control group; Figure 1P,Q,R) showed an overall smooth surface with apparent shallow indentations with no impurities.

### 3.2. Qualitative Assessment of Coated Plates with Bacteria

The SEM analysis of the coated titanium plates with both types of bacteria was undertaken at 5, 10, and 20 k magnification (Figure 2 and Figure 3). PLGA in both the *SA* and *PA* groups was vastly covered by bacteria (Figure 2A,C and Figure 3A,C); with increased magnification at 10 k, the biofilm matrix was noted with bacterial clusters (Figure 2B and Figure 3B). MP in the *SA* group revealed a few bacteria scattered over the plate surface, where the majority of the coated titanium surface was visible (Figure 2D–F). In contrast, in the *PA* group, most of the plate surface was covered by biofilm with very few bacteria (Figure 3D–F). Most of the CH surface was covered with bacteria in both the *PA* and *SA* groups (Figure 2G–I and Figure 3G–I); however, the bacteria were much less dense than in the PLGA group, and the coating layer was visible. MC in the *SA* group had scanty bacteria scattered all over, and the coating layer was also visible (Figure 2J–L); in contrast, the *PA* group surface was covered with biofilm with few bacteria, and the coating layer was substantially covered (Figure 3J–L). MEM in the *SA* group had very few bacteria scattered all over (Figure 2M–O); however, in the *PA* group, the surface was vastly covered in biofilm with few bacteria (Figure 3M–O). In the uncoated plates, both *SA* and *PA* had the most abundant bacteria on the surface, with the coating layer barely visible (Figure 2P–R and Figure 3P–R). 

### 3.3. Qualitative Assessment of the SA and PA Bacteria

Generally, the *SA* bacteria appeared as small, spherical cocci or diplococci dispersed on the plate surface individually or in clusters, with varying amounts in each group (Figure 4), whereas the *PA* bacteria presented as rod-shaped bacilli with varying shapes and sizes, as seen and measured in Figure 5.

### 3.4. Quantitative Assessment of Coated Plates with Bacteria

For the *SA* group, the highest bacterial count was observed in the uncoated plates (1061.00 ± 3.61), whereas the lowest count was detected in MP (18.33 ± 1.16), followed by PLGA, CH, MEM, and MC, respectively, as shown in Table 1 and Figure 2. Overall, the highest bacterial count was observed in the *PA* group with the uncoated plates (2923.33 ± 4.04), whereas the lowest bacterial surface contamination was found in the *PA* group related to CH (2.67 ± 0.58), followed by PLGA, MP, MC, and MEM, respectively, as shown in Table 1 and Figure 3. 

A statistically significant difference was seen between the different materials in the *SA* and *PA* groups (*p* = 0.005, 0.006), respectively (Table 1). Furthermore, a statistically significant difference was noticed in the PLGA-, MP-, and CH-coated plates, regarding *SA* and *PA* (*p* ≤ 0.05); however, there was no statistically significant difference between the *SA* and *PA* groups in relation to the MC- and MEM-coated and uncoated plates (Table 2).

## 4. Discussion

Bacteria were discovered by Leeuwenhoek over three centuries ago. Since then, scientists have continuously attempted to eliminate existing and emerging infectious diseases caused by bacteria and develop antibacterial agents for treatment [19].

To date, the literature lacks information regarding bacterial colonization on titanium microplates used in oral and maxillofacial surgery. However, multiple studies have investigated the microbiological and biofilm characteristics of bacteria on medical devices implanted in the body or on human tissue [20]. 

The prime cause of orthopedic implant-related infections was attributed to *SA*. *SA* was also reported on cardiac implants, vascular grafts, and cosmetic implants [21,22].

Infection is caused primarily by bacterial colonies growing in protected biofilms on the titanium surface. Biofilms are multicellular communities bound together by an extracellular matrix produced by bacteria. The techniques various bacteria employ to create biofilms differ based on environmental factors and strain characteristics [19]. The adherent bacterial cells form multicellular aggregates encased in an extracellular polymeric substance-based matrix [23]. As biofilms are formed over the plates, it is difficult for the antibiotics to enter and clear the infection. Hence, antibiotic therapy is of little use in cases of infection associated with osteosynthesis plates. Archer et al. interpreted that those implanted devices’ surfaces become covered with host proteins that provide a surface for bacterial adhesion and biofilm formation. Once the device becomes infected, it must be completely removed because *SA* biofilm cannot be destroyed by antibiotic treatments [22,24,25]. Biofilms are one of the main factors in chronic infections on the grounds of higher tolerance to antibiotics given that they can combat phagocytosis and the immune system’s other defensive strategies [26]. Therefore, the bacteria within biofilms become resistant to several antibiotics, which drives biofilms to an imminent predicament in therapeutics [27].

Nevertheless, antibiotic tolerance is predominantly dependent on the formation of biofilm; the structure of the extracellular matrix with proteins, lipids, water, glycolipids, polysaccharides, and surfactants; and the architecture of the biofilm, which pertains to how the biomass is organized within the biofilm [28,29]. Consequently, the present investigation was undertaken to evaluate the topographical and ultrastructural architecture of titanium plates coated with PLGA, CH, and/or MEM with or without *SA* or *PA* bacteria.

The overall topographical characteristics of the titanium plate surface showed a relatively uniform and homogeneous coating in each group, with no impact on plate shape or microporosity. The most frequently isolated microorganisms in implant-associated infections are Gram-positive cocci, with *SA* being the most recurrent cause. Gram-negative bacteria such as *PA* are responsible for 10–23% of all cases causing infections. *PA* is considered one of the most difficult-to-treat bacteria because of the growth rate of its multidrug-resistant strains and its ability to develop particular virulence and persistence mechanisms, such as biofilm formation and the production of small colony variants [30].

The highest number of bacteria was found in the uncoated plates associated with *PA*. This is a typical interpretation given the titanium alloy’s lack of innate antibacterial properties and the virulence of *PA* compared to *SA*, resulting in the highest bacterial count among all the groups. As for the PLGA- and CH-coated plates, the resulting bacterial count was substantially lower compared to the negative control group.

The coated plates released MEM into their surroundings immediately after placement to avoid bacterial attachment and biofilm development. Successively, the constant release of bactericidal doses took up to 3 days. To avoid bacterial resistance, all the combined drugs must be released over 3–4 days at the desired site to prevent the initial infection. Undeniably, the release of subinhibitory antibiotic concentrations linked with prolonged treatment time is known to cause bacterial resistance to antimicrobial agents [13,31].

Within the present findings, bacterial colonization was readily observed on the uncoated titanium plates. The highest *SA* bacterial inhibition was related to the MP group, and the CH presented the highest inhibition in the *PA* group. A statistically significant difference was seen between the different materials in the *SA* and *PA* groups. Furthermore, a statistically significant difference was noticed in the PLGA-, MP-, and CH-coated plates regarding *SA* and *PA.* However, there was no statistically significant difference between the *SA* and *PA* groups in relation to the MC- and MEM-coated and uncoated plates.

## 5. Conclusions

The modified titanium plates with polymeric carriers (such as PLGA or CH) supplemented with meropenem seemed to enhance the bacterial inhibition compared to the uncoated plates, reducing the biofilm formation, and hindering bacterial proliferation. Accordingly, if a bacterial biofilm accumulates on the plate’s surface, the presence of the antibacterial agent at the plate site can directly increase its antibacterial efficacy. 

The surface treatment of titanium plates is a challenging, useful approach to avoid infection risks. Further investigations are recommended to acquire new biofunctionalization approaches to improve the antibacterial efficacy of the titanium plates by modifying the polymer properties and, in turn, the antibiotic release profile.

## Figures and Tables

**Figure 1 dentistry-10-00220-f001:**
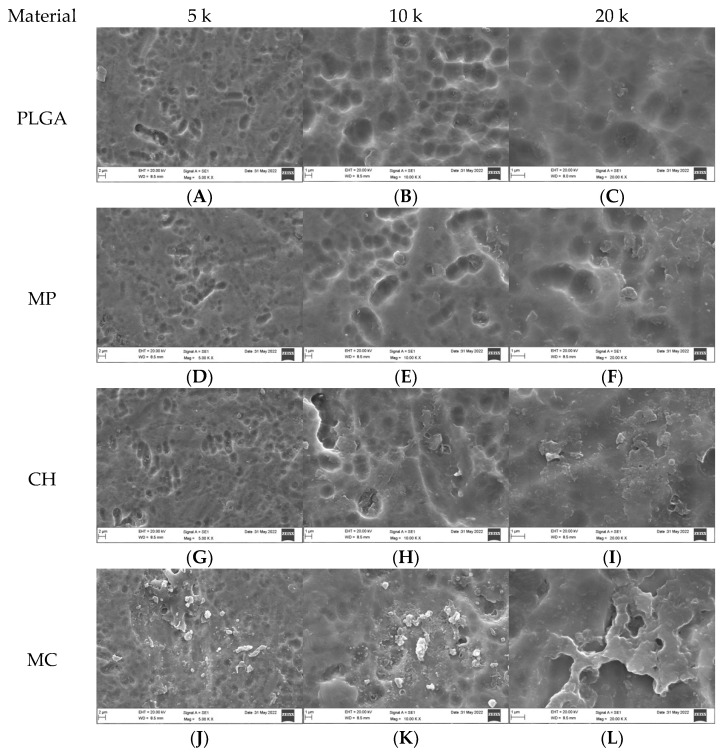
SEM appearance of coated plates without bacteria, including the negative control at 5, 10, and 20k magnification.

**Figure 2 dentistry-10-00220-f002:**
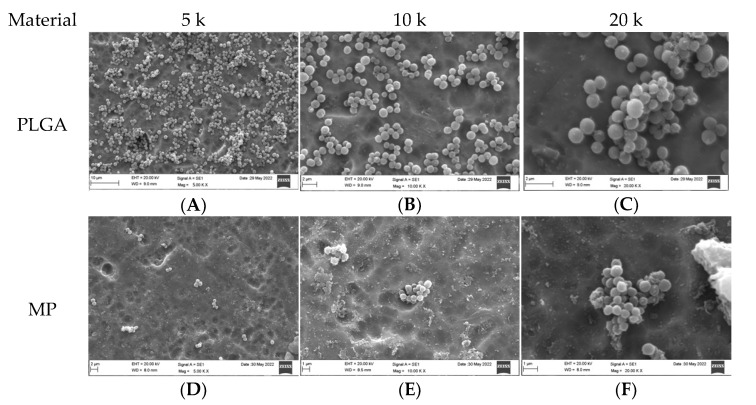
SEM appearance of coated plates with SA bacteria including the negative control. At 5, 10 and 20 k magnification.

**Figure 3 dentistry-10-00220-f003:**
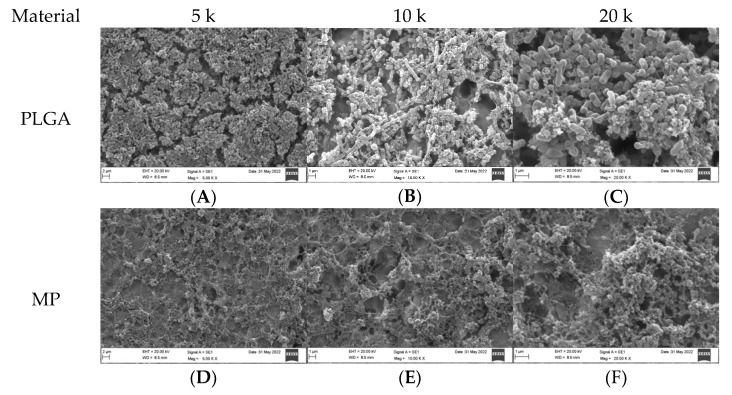
SEM appearance of coated plates with *PA* bacteria including the negative control at 5, 10, and 20 k magnification.

**Figure 4 dentistry-10-00220-f004:**
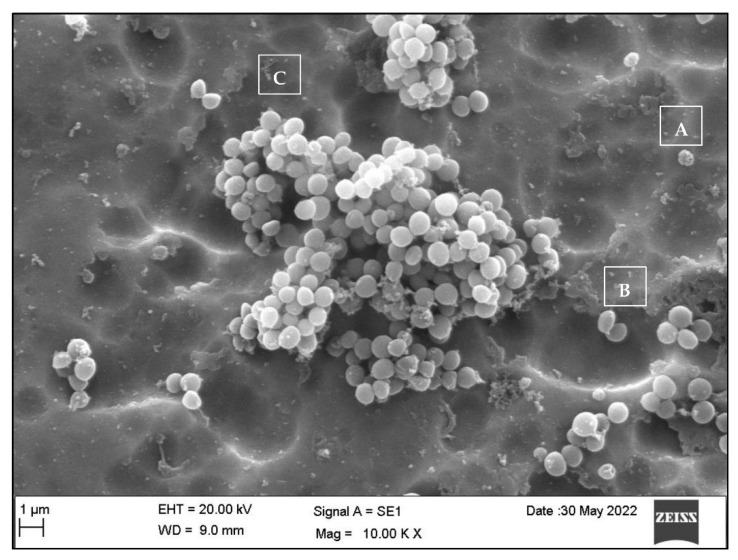
SEM appearance of *SA* as spherical cocci (A), diplococci (B), or in grape-like clusters (C) at 10 k magnification.

**Figure 5 dentistry-10-00220-f005:**
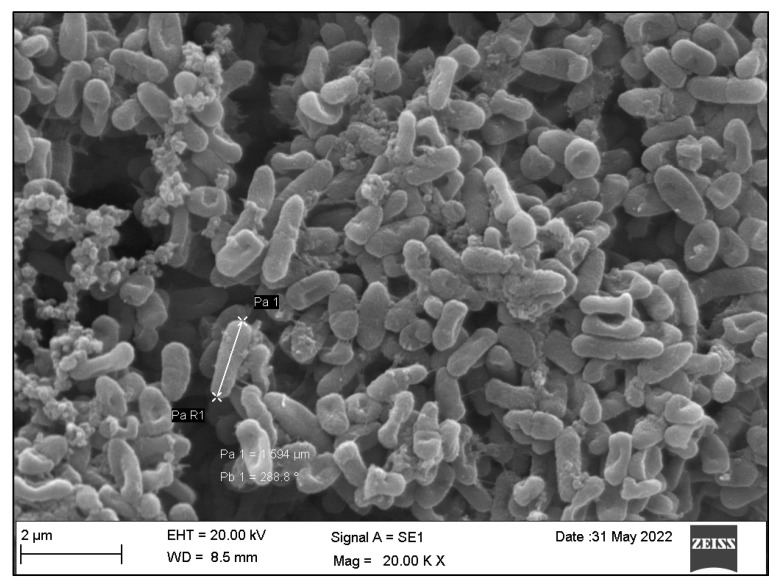
SEM appearance of *PA* as rod-shaped bacilli with varying shapes and sizes at 20 k magnification.

**Table 1 dentistry-10-00220-t001:** Bacterial count on the different materials.

Bacteria	Materials	Mean ± SD	Median	Range	Mean Rank	K.W *p*-Value **
*SA*	PLGA	39.67 ± 1.53	40.00	3.00	5.000	0.005 *
	MP	18.33 ± 1.16	19.00	2.00	2.000	
	CH	119.67 ± 1.16	119.00	2.00	8.000	
	MC	632.00 ± 10.15	634.00	20.00	14.000	
	MEM	440.33 ± 1.53	440.00	3.00	11.000	
	Uncoated	1061.00 ± 3.61	1060.00	7.00	17.000	
*PA*	PLGA	3.33 ± 0.58	3.00	1.00	4.333	0.006 *
	MP	12.67 ± 1.16	12.00	2.00	8.000	
	CH	2.67 ± 0.58	3.00	1.00	2.667	
	MC	1261.67 ± 6.81	1264.00	13.00	11.000	
	MEM	2108.00 ± 8.54	2107.00	17.00	14.000	
	Uncoated	2923.33 ± 4.04	2924.00	8.00	17.000	

* Statistically significant at *p* ≤ 0.05. ** Kruskal–Wallis *p* value.

**Table 2 dentistry-10-00220-t002:** Bacterial count within the different materials.

Compound	Bacteria	Sum of Ranks	*p*-value
PLGA	*SA*	15.00	0.04 *
	*PA*	6.00	
MP	*SA*	15.00	0.04 *
	*PA*	6.00	
CH	*SA*	15.00	0.04 *
	*PA*	6.00	
MC	*SA*	6.00	0.05
	*PA*	15.00	
MEM	*SA*	6.00	1.05
	*PA*	15.00	
Uncoated	*SA*	6.00	0.05
	*PA*	15.00	

* Statistically significant at *p* ≤ 0.05.

## Data Availability

The data supporting the reported results analyzed or generated during the study are available upon request from the corresponding author.

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
