# Peer review of "Topographical and Ultrastructural Evaluation of Titanium Plates Coated with PLGA, Chitosan, and/or Meropenem: An In Vitro Study"

_dentistry, 2022, doi:10.3390/dj10120220_

Round 1

Reviewer 1 Report

In this study, the antibacterial properties of coatings of PLGA, CH, PLGA and CH loaded with MEM on titanium were investigated.  Some comments were given in order to improve the manuscript:

Materials and Methods

1. Were there any surface pre-treatment of titanium plates, e.g., polishing and then rinsing in distilled water?

2. Did you examine the chemical composition of different coatings by EDX?

3. The coating is 1% w/v in acetone for each group. Please check solution 3 as only 5 mg meropenem in 2.5 ml acetone.

4. For solutions 1 and 2, the total amount (PLGA + meropenem) = 30 mg in 2.5 ml acetone. Please check if it is 20 mg PLGA and 5 mg meropenem in 2.5 mL acetone. That’s (20 + 5) / 2.5 x 100% = 1% w/v.     

5. What is the total number of titanium plates used in this study? What is the sample size for each group used for antibacterial test and SEM examination?

 Results

1. For Figures 1 – 5, scale bars are missed.

 Discussion

1. The polylactic-co-glycolic acid coating on titanium exhibited antibacterial activities against SA and PA which is comparable to MP coating. It is expected that polylactic-co-glycolic acid coating exhibited little antibacterial activity.  

2. It is surprised that the antibacterial activities of chitosan loaded with meropenem coating against SA and PA are significantly lower than chitosan coating.    

3. From the results, the two bacteria strains, SA and PA, developed drug resistance against the antibiotic, meropenem (MEM),

 Conclusion

1. “Coating titanium plates with PLGA or CH with MEM appeared to enhance bacterial inhibition over uncoated plates”. I may suggest to re-write it as “Coating titanium plates with PLGA and CH enhances bacterial inhibition over uncoated plates”. As the two bacteria strains developed drug resistance against MEM.  

Reviewer 2 Report

The manuscript describes an evaluation of the topographical and ultrastructural architecture of titanium plates coated with Polylactic Co-Glycolic Acid (PLGA), chitosan (CH), and/or meropenem (MEM) with or without Staphylococcus aureus (SA) or Pseudomonas aeruginosa (PA) bacteria.

The scope is moderate; the method is appropriate for achieving the objectives; the impact is moderate. The novelty is not apparent. Please write a novelty and significance statement upfront. The manuscript merits informative images and sufficient level of English usage.

However, the scale bars are missing in all of the SEM images. Although titanium plates are introduced in the first paragraph, nothing has been mentioned about their intrinsic properties facilitating their usage.

Hence, minor revision is recommended.

1. L37, here is essential to state why titanium is a better replacement of the other metals, can state "Ti and Ti‐containing alloys have an extreme strength‐to‐weight ratio, high corrosion resistance, biocompatibility and inherent ability to osseointegrate, making them ideal for biomedical applications. citing [10.3390/met12030406]

2. L49, please state their advantages.

3. L108, why is the time scale 24 hours and not 48 or 72 hours?

4. All SEM images need scale bars.

5. Tables 1 and 2 have the same generic titles. Please rewrite to reflect the contents and the difference between the 2.

6. Although the title mentions “Ultrastructural Evaluation” which part of the content is about ultrastructural evaluation is not clear. And Ultrastructural Evaluation is never mentioned again in the main text. Please clarify.

Reviewer 3 Report

Good work and a well-written manuscript.

The only part need revision is the conclusion. It seems very confident that the tested coated plates are bacterial resistant. Even though the results showed significant differences between the coated groups compared to the control, the way the biofilm study is handled gives us an overall assessment with many variables compared to in vivo studies, even for the same bacteria and substrate. I would rather revise the conclusion to be more neutral. 
